# A Poly-D-Mannose Synthesized by a One-Pot Method Exhibits Anti-Biofilm, Antioxidant, and Anti-Inflammatory Properties In Vitro

**DOI:** 10.3390/antiox12081579

**Published:** 2023-08-08

**Authors:** Dandan Tian, Yu Qiao, Qing Peng, Yuwei Zhang, Yuxin Gong, Linbo Shi, Xiaoyan Xiong, Mengxin He, Xiaoqing Xu, Bo Shi

**Affiliations:** Feed Research Institute, Chinese Academy of Agricultural Sciences, Beijing 100081, China; 82101201074@caas.cn (D.T.); qiaoyu@caas.cn (Y.Q.); pengqing@caas.cn (Q.P.); zhangyuwei2527@163.com (Y.Z.); 82101211095@caas.cn (Y.G.); 82101202150@caas.cn (L.S.); 82101215432@caas.cn (X.X.); 82101212159@caas.cn (M.H.)

**Keywords:** poly-D-mannose, anti-biofilm, antioxidant, anti-inflammatory

## Abstract

In this study, D-mannose was used to synthesize poly-D-mannose using a one-pot method. The molecular weight, degree of branching, monosaccharide composition, total sugar content, and infrared spectrum were determined. In addition, we evaluated the safety and bioactivity of poly-D-mannose including anti-pathogen biofilm, antioxidant, and anti-inflammatory activity. The results showed that poly-D-mannose was a mixture of four components with different molecular weights. The molecular weight of the first three components was larger than 410,000 Da, and that of the fourth was 3884 Da. The branching degree of poly-D-mannose was 0.53. The total sugar content was 97.70%, and the monosaccharide was composed only of mannose. The infrared spectra showed that poly-D-mannose possessed characteristic groups of polysaccharides. Poly-D-mannose showed no cytotoxicity or hemolytic activity at the concentration range from 0.125 mg/mL to 8 mg/mL. In addition, poly-D-mannose had the best inhibition effect on *Salmonella typhimurium* at the concentration of 2 mg/mL (68.0% ± 3.9%). The inhibition effect on *Escherichia coli* O157:H7 was not obvious, and the biofilm was reduced by 37.6% ± 2.9% at 2 mg/mL. For *Staphylococcus aureus* and *Bacillus cereus*, poly-D-mannose had no effect on biofilms at low concentration; however, 2 mg/mL of poly-D-mannose showed inhibition rates of 33.7% ± 6.4% and 47.5% ± 4%, respectively. Poly-D-mannose showed different scavenging ability on free radicals. It showed the best scavenging effect on DPPH, with the highest scavenging rate of 74.0% ± 2.8%, followed by hydroxyl radicals, with the scavenging rate of 36.5% ± 1.6%; the scavenging rates of superoxide anion radicals and ABTS radicals were the lowest, at only 10.1% ± 2.1% and 16.3% ± 0.9%, respectively. In lipopolysaccharide (LPS)-stimulated macrophages, poly-D-mannose decreased the secretion of nitric oxide (NO) and reactive oxygen species (ROS), and down-regulated the expression of tumor necrosis factor-alpha (TNF-α) and interleukin-6 (IL-6). Therefore, it can be concluded that poly-D-mannose prepared in this research is safe and has certain biological activity. Meanwhile, it provides a new idea for the development of novel prebiotics for food and feed industries or active ingredients used for pharmaceutical production in the future.

## 1. Introduction

Mannan is widely found in nature and is one of the main components of hemicelluloses in plant cell walls [1]. Mannan includes pure mannan composed only of mannose, as well as glucomannan, galactomannan, and galactoglucomannan based on carbohydrates or acid substitutions in the backbone [2,3]. In addition, fungi are also important sources of mannan, including some mushrooms and yeast. The proportion of mannan in the yeast cell wall is up to 31% and has been extensively studied [4,5].

Mannan shows various functional properties by improving the overall health of humans and animals, and has thus become a popular research topic. As a plant-derived polysaccharide, Konjac glucomannan can regulate the distribution of microflora in constipated mice by increasing the abundance of *Spirillomyces*, *Ruminococcus*, and *Helicobacter* in the intestinal tract of colitis mice, and down-regulating the abundance of *Bacteroides* and *Sartella*, improving the intestinal metabolic environment and microbial imbalance [6]. Some other studies have confirmed that yeast mannan can be utilized by intestinal bacteria to increase the richness and diversity of intestinal flora and produce SCFAs [7]. Yeast mannan is always used as an antibiotic alternative in the feed industry because it can act as a binding site for pathogenic lectin to inhibit the adhesion of pathogenic bacteria to intestinal cells [8]. Furthermore, mannan can improve the activity of antioxidant enzymes such as superoxide dismutase, catalase, and glutathione peroxidase, increase the scavenging ability of hydroxyl free radicals and superoxide anions, enhance the anti-lipid peroxidation effect, and improve the body’s antioxidant capacity [9]. Patchaiyappan et al. [10] indicated that galactomannan extracted from the seeds of Strychnos potatorum was effective in scavenging DPPH and hydroxyl radicals, which was concentration dependent. Reports also showed yeast mannan had antioxidant properties, such as the ability to scavenge superoxide anions and hydroxyl radicals [11]. In addition, immune and anti-inflammatory activities are also important functional activities of mannan. Macrophages treated with mannan can increase the production of TNF-α by activating the CD14 receptor and toll-like receptor-4 (TLR-4) [12]. Meanwhile, mannan can significantly reduce the levels of some inflammatory factors such as interleukin 10 (IL-10), interleukin 4 (IL-4), and TNF-α in inflammation in experimental animals [13]. However, few studies have reported the antibacterial and anti-biofilm activities of mannan. Smith et al. [14] showed that mannan-oligosaccharides extracted from the yeast cell wall can act as an alternative to antibiotic growth promoter to play an antibacterial role. Lakra et al. [15] found that an exopolysaccharide from Weissella confusa MD1 was mannan and showed excellent anti-biofilm activity against *Staphylococcus aureus*, *Listeria monocytogenes*, *Salmonella enterica*, and *Salmonella typhi*.

At present, mannan are mostly obtained by isolation from the plant or yeast cell wall by water extraction; enzymatic, acid/alkali hydrolysis, or physical methods [16,17]; synthesis by an engineering strain [18]; or enzymatic synthesis [19] in vitro. However, some problems still exist, such as high cost, complex purification process, degradation of the strain, and low enzyme activity [20]. In addition, the purity of mannan obtained by these methods is always unsatisfactory, and the monosaccharide composition is also different, resulting in an unpredictable structure and thus unstable effects in animals. The one-pot method is a common method for the synthesis of branched polymers and is also used for the synthesis of branched polysaccharides [21,22]. Compared with the traditional chemical synthesis method, the one-pot method does not need complicated synthesis steps and the whole reaction process is carried out in a reaction system. The product of the last reaction can be used as the substrate of the next reaction, and the intermediate can be prepared without separation and purification [23]. Therefore, this method greatly simplifies the operation steps and reduces the cost.

In this study, poly-D-mannose was synthesized using the one-pot method under the condition of high temperature and negative pressure using D-mannose. The molecular weight, degree of branching, monosaccharide composition, and total sugar content were further determined, and the characteristic groups were analyzed by Fourier transform infrared spectroscopy (FT-IR). A scanning electron microscope (SEM) was used to observe the surface morphology. The safety of poly-D-mannose, including its cytotoxicity and hemolysis, was also evaluated. Further, we carried out relevant experiments on its functional activities such as anti-biofilm, antioxidant, and immune/anti-inflammatory activities.

## 2. Materials and Methods

### 2.1. Materials

*Salmonella enterica* subsp. *Enterica serovar Typhimurium* (CICC 22956/ATCC 14028), *Escherichia coli* O157:H7 (CICC 10907), and *Bacillus cereus* (CICC 21261) were purchased from China Center of Industrial Culture Collection (CICC). *Staphylococcus aureus* (CGMCC 1.291) was purchased from the China General Microbiological Culture Collection Center (CGMCC). RAW264.7 macrophages were purchased from Beijing Dingguo Changsheng Biotechnology Co., Ltd., Beijing, China. Vero cells were purchased from Procell Life Science & Technology Co., Ltd., Wuhan, China. Dextran standards were obtained from Sigma-Aldrich (São Paulo, Brazil). The other chemicals used in this study were analytical grade and purchased from Solarbio Science &Technology Co., Ltd., Beijing, China and Thermo Fisher Scientific (Waltham, MA, USA).

### 2.2. Preparation of Poly-D-Mannose

The synthesis of poly-D-mannose followed the preparation method of polydextrose [24,25]. In detail, 30 g D-mannose was mixed with 0.9 g solid citric acid, and then reacted in a 1 L glass flask for 30 min at 155 °C, 120 rpm, and −0.1 Mpa for the polymerization of D-mannose monomer. During the reaction, water vapor was extracted and collected by negative pressure and condensation. After the reaction, the products were suspended in 100 mL deionized water under the condition of room temperature. The solution was dialyzed for 4 days against deionized water at room temperature with a dialysis bag (MWCO 500 Da, Biotopped, Beijing, China) and poly-D-mannose was obtained after lyophilization.

### 2.3. SEM Observation

The microstructure of poly-D-mannose was observed using an SEM (SU8000, Hitachi, Tokyo, Japan). Before the observation, the powder samples were sprayed with a layer of gold and tested at a voltage of 3 KV.

### 2.4. FT-IR Analysis

The potassium bromide pellet pressing method was used for FT-IR spectroscopy (Vertex 70V; Hyperion 2000; Bruker, Berlin, Germany). The functional groups, molecular structure, and chemical composition of poly-D-mannose were analyzed according to the results of the infrared spectra over a wavelength range of 4000 cm^−1^ to 400 cm^−1^. The D-mannose and citric acid were measured as controls. Samples were measured three times in parallel.

### 2.5. Molecular Weight Determination

The molecular weight of poly-D-mannose was determined using the method of water-based size exclusion chromatography [26,27]. Briefly, a Waters 2695 HPLC system (Milford, MA, USA) equipped with a TSK gel GMPW XL column (300 mm × 7.8 mm, Tosoh Corp., Tokyo, Japan) and a refractive index detector were used. The analysis was performed at 35 °C with a mobile phase of deionized water at a flow rate of 0.8 mL/min. Dextran with different molecular weights was used as the standard. The standard curve (y = −0.7873x + 13.555, R^2^ = 0.9903) was made with the logarithm of the molecular weight of dextran as the vertical coordinate (y) and the retention time as the horizontal coordinate (x). The sample was measured three times in parallel.

### 2.6. Total Sugar Content Determination

The phenol-sulfuric acid method was used to determine total sugar content [28]. D-mannose was used as a standard. Three parallel treatments were performed for samples. The standard curve was made (y = 6.9229x + 0.0739, R^2^ = 0.9909) with D-mannose mass as the horizontal coordinate (x) and absorbance at 490 nm as the vertical coordinate (y).

### 2.7. Monosaccharide Composition Determination

Monosaccharide of poly-D-mannose was obtained after hydrolysis with sulfuric acid according to Xu et al. [28]. Next, the monosaccharide composition was determined by high-performance liquid chromatography (HPLC) equipped with a Shodex HILICpak VG-50 4E (250 mm × 4.6 mm, Shodex, Tokyo, Japan) and an evaporative light-scattering detector (ELSD) (6100 Chromachem, ESA Inc., Marthasville, MO, USA). The detection was performed at 40 °C with a mobile phase of acetonitrile: methanol: water (77.5:15:7.5) at a flow rate of 1 mL/min. Mannose and glucose were used as monosaccharide standards. Three parallels were included in each sample.

### 2.8. Determination of Branching Degree

According to the method described by Wang et al. [29], the number of terminal residues, branched residues, and linear residues in poly-D-mannose was determined by methylation analysis. Ten milligrams of dried poly-D-mannose were dissolved with 1.5 mL dimethyl sulfoxide (DMSO), and 80 mg sodium hydroxide was added, followed by stirring for 2 h. Iodomethane (1.5 mL) was added to the reaction solution and reacted in darkness for 2 h, and 3 mL of deionized water was added to terminate the reaction. The reaction solution was dialyzed in a 1KDa dialysis bag for 72 h and then freeze-dried. The above steps were repeated three times. Two milliliters of 2 M trifluoroacetic acid were added to the methylated sample and hydrolyzed at 121 °C for 2 h. The reaction solution was dried to neutral under reduced pressure. Then, 1 mL of 4% sodium borohydride was added and left at room temperature for 1.5 h. Glacial acetic acid was added to neutralize the reaction solution by drops and the dried. Next, 2 mL of 99.5% methanol solution was added and dried under pressure, with repeating 3–4 times. One milliliter pyridine and 1 mL acetic anhydride were added to the sample in turn, reacted overnight, and rotated to dry at 80 °C. The above samples were extracted by dichloromethane to obtain Partially Methylated Alditol Acetates (PMAAs) for Gas Chromatography-Mass Spectrometry (GC-MS) analysis. The GC was equipped with a RTX-5MS column (30 m × 0.25 mm × 0.25 μm). The temperature program was 140 °C for 2 min, then 2 °C/min to 250 °C. Helium at a flow rate of 1 mL/min was used as carrier gas. The MS was operated with an electron impact mode at 350 V. EI was used as the ionization mode and the ion source temperature was 210 °C. The sample was measured three times in parallel.

Degree of branching (DB) was calculated by the method of Hawker et al. [30], as shown in the following formula:$$\mathrm{DB}=\frac{N_{T}+N_{B}}{N_{T}+N_{B}+N_{L}}\times 100\%$$
where DB is the degree of branching and N_T_, N_B_, and N_L_ are the numbers of terminal residues, branch residues, and linear residues, respectively.

### 2.9. NMR Spectroscopy

Twenty-five milligrams of poly-D-mannose were exchanged with 0.5 mL of D_2_O (99.96%) two consecutive times. After dissolving with 0.5 mL of D_2_O (99.96%), ^1^H and ^13^C were determined using a Bruker Avance-500 NMR spectrometer (Bruker Corporation, Karlsruhe, Germany). DMSO was used as an internal standard.

### 2.10. In Vitro Evaluation of Poly-D-Mannose Safety

#### 2.10.1. Cell Cytotoxicity Assay

Cell cytotoxicity of poly-D-mannose was assessed using Vero cells according to the method described by Rubini et al. [31], with slight modifications. The detailed operations were as follows. The Vero cells were cultured in Dulbecco Modified Eagle’s medium (DMEM) supplemented with 10% fetal bovine serum (FBS) and 1% penicillin-streptomycin (P/S) at 37 °C in a humid condition with 5% CO_2_. Cells were seeded in a 96-well plate (Costar, Corning, NY, USA) with the density of 1 × 10^4^ cells/well and 100 µL/well. After growth for 24 h at 37 °C under the condition of 5% CO_2_ and certain humidity, the supernatant was removed. Cells were treated with different concentrations of poly-D-mannose (0.125, 0.25, 0.5, 1, 2, 4, 8 mg/mL) prepared with DMEM medium. Three parallels were performed for each sample. After 24 h of treatment, 30 µL 3-(4,5-dimethyl-2-thiazolyl)-2,5-diphenyl-2-H-tetrazolium bromide (MTT) (5 mg/mL) was added and incubated at 37 °C for 4 h. One hundred microliters of DMSO were added to each well and mixed thoroughly. Optical density at 540 nm was measured and the percentage of cell survival was calculated.

#### 2.10.2. Hemolysis Capacity

Hemolysis assay of poly-D-mannose was carried out based on the method of Upadhyay et al. [32] with minor modifications. Fresh sterile defibrillated sheep blood was centrifuged at 10,000× *g* for 5 min after mixing with sterile PBS at 1:2 (*v*/*v*). The supernatant was discarded and the erythrocytes were washed 3–4 times with sterile PBS until the supernatant became clear and transparent. Erythrocyte suspension (10%) was obtained after resuspending in 20 mL PBS. A mixture of 0.2 mL erythrocyte suspension and 0.8 mL sample (0.125–8 mg/mL) was reacted at 37 °C for 4 h and 20 h, respectively, and centrifuged at 10,000× *g* for 5 min. Deionized water and 0.2% TritonX-100 were used as different positive controls, and PBS was used as the negative control. Three parallels were performed for each sample. The reaction supernatant (100 µL) was added to the 96-well plate and optical density was determined at 577 nm. Hemolysis rate (%) = (OD_sample_ − OD_negative_)/(OD_positive_ − OD_negative_). Judgment of non-hemolytic was less than 5% of hemolysis rate.

### 2.11. Biological Activity of Poly-D-Mannose

#### 2.11.1. Anti-Biofilm Activity Assay

Ninety microliters (10^6^ CFU/mL) of bacterial suspension (*B. cereus*, *Staphylococcus aureus*, *S.* Typhimurium and *E. coli* O157:H7) were added to a 96-well plate, and then 10 µL poly-D-mannose dissolved in sterile PBS was added with the final concentrations of 0.25, 0.5, 1, and 2 mg/mL, respectively. Each group was performed in quadruplicate. The treatment with bacterial suspension and sterile PBS was the negative control, and the treatment with bacterial culture medium and samples was the blank control. Ninety-six-well plates were placed in an incubator without shaking at 37 °C for 24 h.

The culture medium in each well was discarded and washed four times with 120 µL sterile PBS buffer to remove the unattached cells. The biofilm in the 96-well plate was stained with MTT and incubated in the dark for 3 h at 37 °C [28]. After incubation, the MTT solution was removed and 100 µL DMSO was added to each well to oscillate at a slow speed for 10 min for rinsing. The absorbance at 490 nm was read using a Microplate Reader (BioTek, Winooski, VT, USA). The biofilm formation of each strain was expressed by the following formula: (OD_sample_/OD_negative_) × 100%.

#### 2.11.2. Antioxidant Activity Assay

##### DPPH Radical Scavenging Activity

The DPPH radical scavenging ability of poly-D-mannose was performed according to Xie et al. [33]. Details were as follows: 100 μL poly-D-mannose samples with different concentrations (0.125–8 mg/mL) and 100 μL DPPH (100 μM, dissolved in methanol) were added to 96-well plates in turn, mixed thoroughly, and incubated in the dark at room temperature for 30 min. The absorbance was measured at 517 nm. Three parallels were performed for each treatment. The deionized water was used as the negative control and different concentrations of ascorbic acid solution were used as the positive control. DPPH scavenging activity (%) = [1 − (A_S_ − A_S0_)/(A_C_ − A_C0_)] × 100%, where A_S_: the absorbance of different concentrations of sample; A_S0_: the background absorbance of sample without DPPH; A_C_: the control absorbance without sample; A_C0_: the background absorbance without sample and DPPH.

##### Hydroxyl Radical Scavenging Activity

The hydroxyl radical scavenging capacity was carried out in accordance with Wu et al. [34]. Fifty microliters of PBS (20 mM, pH7.4), 25 µL 1,10-phenylene (2.5 mM), 25 µL FeSO_4_ (2.5 mM), and 25 µL H_2_O_2_ (20 mM) were successively added to a 96-well plate and thoroughly mixed, and 100 µL poly-D-mannose of different concentrations (0.125–8 mg/mL) was added to the 96-well plate. Ascorbic acid of the same concentration was used as the positive control, and deionized water was used as the negative control. Three parallels were performed for each treatment. The absorbance of the reaction solution was measured at 536 nm. Hydroxyl radical scavenging activity (%) = (A_S_ − A_C_)/(A_0_ − A_C_) × 100%, where A_S_: the absorbance of different concentrations of sample; A_C_: the control absorbance without sample; A_0_: the background absorbance without sample and H_2_O_2_.

##### Superoxide Anion Scavenging Activity

The scavenging activity of poly-D-mannose on superoxide radicals was determined by pyrogallic acid, following the reported methods [35]. Fifty microliters of Tris-HCl buffer (pH = 8.0, 150 mM) and 25 µL of pyrogallic acid (1.5 mM, dissolved in 10 mM HCl) were mixed; then, 100 µL of poly-D-mannose solution of different concentrations (0.125–8 mg/mL) was added. The solution stood at room temperature for 30 min. Ascorbic acid of the same concentration was used as the positive control and deionized water was used as the negative control. Three parallels were performed for each treatment. The absorbance of the reaction solution was measured at 325 nm. Superoxide anion scavenging activity (%) = [1 − (A_S_ − A_S0_)/(A_C_ − A_C0_)] × 100%, where A_S_: the absorbance of different concentrations of sample; A_S0_: the background absorbance of sample without pyrogallic acid; A_C_: the control absorbance without sample; A_C0_: the background absorbance without sample and pyrogallic acid.

##### ABTS Scavenging Activity

The ABTS radical scavenging activity was determined according to the method reported by Gu et al. [36]. Five milliliters and 7 mM of ABTS stock solution were mixed with 88 μL, 140 mM potassium persulfate, and placed in dark overnight to obtain an ABTS working solution. The ABTS working solution was diluted with deionized water to adjust the OD_734_ = 0.7 ± 0.02 for the following measurement. Subsequently, 100 μL poly-D-mannose (0.125–8 mg/mL) with different concentrations was mixed with 100 μL ABTS working solution, and then placed in the dark for 6 min at room temperature. Deionized water was used as the negative control and the ascorbic acid was used as the positive control. Three parallels were performed for each treatment. The absorbance of the reaction solution was measured at 734 nm. ABTS scavenging activity (%) = [1 − (A_S_ − A_S0_)/(A_C_ − A_C0_)] × 100%, where A_S_: the absorbance of different concentrations of sample; A_S0_: the background absorbance of sample without ABTS; A_C_: the control absorbance without sample; A_C0_: the background absorbance without sample and ABTS.

#### 2.11.3. Anti-Inflammatory Activity

##### Culture of RAW264.7 Macrophages

Macrophages were inoculated in the petri dishes with about 7 mL DMEM medium containing 10% FBS and 1% P/S solution. Cells were grown in a humidified atmosphere with 5% CO_2_ at 37 °C.

##### Cell Proliferation, NO, and Cytokine (IL-6 and TNF-α) Determination

Macrophages were inoculated in 96-well plates with 200 µL medium per well for 24 h with the density of 5 × 10^4^ cells/well. Poly-D-mannose of different concentrations (0.125, 0.25, 0.5, 1, 2 mg/mL) was prepared in DMEM medium supplemented with 1 μg/mL LPS. Macrophages were treated for 24 h. Each concentration was performed in quadruplicate. After 24 h, the supernatant of the culture medium was collected for the determination of NO and cytokines, and the cells in the pore plate were used for the determination of cell proliferation using 3-(4,5-dimethylthiazol-2-yl)-5-(3-carboxymethoxyphenyl)-2-(4-sulfophenyl)-2H-tetrazolium (MTS) assay. MTS solution was prepared with DPBS buffer (0.2 g/L KCl, 8.0 g/L NaCl, 0.2 g/L KH_2_PO_4_, and 1.15 g/L Na_2_HPO_4_, adjust pH = 7.35 with 1 M HCl or NaOH, add 100 mg/L MgCl_2_ and 133 mg/L CaCl_2_•2H_2_O); 250 mg of MTS was added to 125 mL of DPBS, mixed well, adjusted pH = 6–6.5, filtered, and stored at −20 °C. Before usage, 5% phenazine methosulfate (PMS) solution was added to MTS solution. To a 96-well plate with supernatant removed, 100 µL MTS solution was added (100 µL/well), incubated at 37 °C for 3 h, and the absorbance at 490 nm was measured. Cell activity was calculated using the following formula: (OD_sample_/OD_negative_) × 100%.

Griess reagent (A: 1% sulfanilic acid in 5% phosphoric acid; B: 0.1% N-(1-napthyl)-ethylenediamine dihydrochloride; mix A and B just before using) was used to determine the content of NO. The specific methods were as follows: 100 µL cell supernatant and different concentrations (0, 3.125, 6.25, 12.5, 25, 50, 100 µM) of sodium nitrite (NaNO_2_) were mixed with 100 µL Griess reagent in a 96-well plate, the mixtures were incubated at room temperature for 5 min, and the absorbance at 550 nm was measured. Each group was performed in triplicate. The content of NO in cell supernatant was calculated according to the standard curve (y = 0.0091x + 0.0505, R^2^ = 0.9995; x: the concentration of NaNO_2_, y: the OD of NaNO_2_).

ELISA was used to determine the content of cytokines (IL-6 and TNF-α) in cell supernatant. The detailed process was performed according to the instructions in the ELISA kit (BioLegend, San Diego, CA, USA). IL-6 and TNF-α of different concentrations (0, 7.8125, 15.625, 31.25, 62.5, 125, 250, and 500 pg/mL) were used as the standards, and the cell supernatant was used for the test after being diluted 100 times. The standard curves of IL-6 (y = 1.5311x + 2.2325, R^2^ = 0.997; x: logarithm of OD_450_, y: logarithm of standard concentration) and TNF-α (y = 1.7469x + 1.79, R^2^ = 0.9736; x: logarithm of OD_450_, y: logarithm of standard concentration) were obtained according to measuring OD_450_. The contents of IL-6 and TNF-α in the cell supernatant were obtained according to the standard curve.

##### ROS Determination

Cells (5 × 10^4^ cells/well) were inoculated into the 96-well plate with a black wall and transparent bottom (Corning, Costar, Corning, NY, USA) and grown for 24 h. After the supernatant was discarded, cells were treated with poly-D-mannose solution with different concentrations (0.125, 0.25, 0.5, 1, 2 mg/mL) prepared with DMEM medium supplemented with 1 μg/mL LPS for 24 h. DMEM with and without 1 μg/mL LPS was used as positive and negative controls, respectively. The culture medium without cells was used as the blank control. Each treatment was performed in quadruplicate. After removed the supernatant, the adherent cells were washed twice with cold PBS under the condition of an ice bath. Two hundred microliters of 2′,7′-dichlorofluorescein diacetate (DCFH-DA) (10 µM) were added to each well and incubated for 30 min at 37 °C, 5% CO_2_ constant humidity in an incubator. The fluorescence intensity of each well was measured by a microplate detector (excited light at 485 nm and emitted light at 528 nm). The dye solution in the pore plate was removed, and the cells were washed twice with cold PBS under the condition of an ice bath. The fluorescence intensity of treated cells was also observed through a fluorescence microscope (20×) under a green fluorescent protein channel.

### 2.12. Statistical Analysis

All statistical analyses were conducted using SPSS version 17.0 (SPSS Inc., Chicago, IL, USA). One-way ANOVA with Duncan’s test was used for multiple comparisons of samples of different concentrations. Data in this study are presented as mean ± SD. *p* < 0.05 was considered statistically significant.

## 3. Results

### 3.1. Characterization of Poly-D-Mannose

#### 3.1.1. SEM Observation

The microstructures and microscopic shape of poly-D-mannose were observed using SEM, and the images are shown in Figure 1. Under the SEM, poly-D-mannose presented an irregular lamellar structure, with granular debris on the lamellar surface. Furthermore, some sheet surfaces have convex parallel structures.

#### 3.1.2. FT-IR

The FT-IR spectrum of poly-D-mannose is shown in Figure 2a. Compared with the infrared spectra of D-mannose and citric acid, poly-D-mannose showed some characteristic peaks. Some typical absorption peaks of the polysaccharides in the range of 4000–400 cm^−1^ were found. The FT-IR spectrum of poly-D-mannose showed that a strong absorption peak around 3473 cm^−1^ was considered to be the characteristic absorption of the O-H [37]. The peak at 2946 cm^−1^ represented the asymmetric stretching vibration of C-H [38], and the peak around 1664 cm^−1^ was due to the asymmetric stretching vibration of C = O [36]. In addition, there was a distinct absorption peak at 1097 cm^−1^, which was caused by the stretching vibration of the C-O side groups and the vibration of the C-O-C glycosidic bond indicating the presence of pyranose units [39]. The absorption peaks at 821 cm^−1^ and 921 cm^−1^ belong to α-type and β-type glycosidic bonds, respectively [40]. Typical absorption peaks of poly-D-mannose in FT-IR suggested its polysaccharide characteristics, indicating the feasibility of the method.

#### 3.1.3. Molecular Weight

The molecular weight distribution of the poly-D-mannose is shown in Figure 2b. The results showed that the poly-D-mannose was a compound with four components. The molecular weight of the first three components was larger than 410,000 Da, which was beyond the range of the standard curve. The molecular weight of the fourth was 3884 Da.

#### 3.1.4. Total Sugar Content and Monosaccharide Composition

The total sugar content of poly-D-mannose was 97.70% according to the standard curve, suggesting high purity. The HPLC chromatogram in Figure 2c shows that the poly-D-mannose was a homopolysaccharide that only consisted of D-mannose.

#### 3.1.5. Degree of Branching

The types of sugar residues of poly-D-mannose are displayed in Table 1. Poly-D-mannose had three types of sugar residues, namely, terminal, branched, and linear residues. The relative molar percentage was 29.39, 23.71, and 46.90, respectively. The branching degree of poly-D-mannose was calculated to be 0.53, which indicated that poly-D-mannose was a highly branched glycan.

#### 3.1.6. NMR Analysis

^1^H and ^13^C NMR spectra are shown in Figure 3. In the ^1^H NMR spectra, the signal of polysaccharide is mostly concentrated in the range of 3.0–5.5 ppm. The signal between 4.8–5.2 ppm is the proton signal on anomeric carbons. The chemical shift of protons on α-pyranose anomeric carbon is greater than 5.0 ppm, while the chemical shift of protons on β-pyranose anomeric carbon is less than 5.0 ppm [41]. In Figure 3a, four types of anomeric protons were found: δ 5.14 ppm, δ 5.00 ppm, δ 4.95 ppm, and δ 4.81 ppm. It can be inferred that poly-D-mannose contains both α and β glycosidic bond types. In the ^13^C NMR spectrum (Figure 3b), five peaks were observed in the range of δ 95–110 ppm: δ 107.58 ppm, δ 105.28 ppm, δ 102.18 ppm, δ 100.56 ppm, and δ 99.75 ppm. A chemical shift in the range of 95–103 ppm indicates that the anomeric carbon belongs to the α-type glycosidic bond, while a shift in the range of 103–110 ppm indicates a β-type glycosidic bond [42]. Therefore, according to the results of the ^13^C spectrum, it can also be concluded that poly-D-mannose has both α and β types of glycosidic bonds. The NMR results were consistent with FT-IR a nalysis.

### 3.2. Safety Profile

#### 3.2.1. Cytotoxicity

The effect of poly-D-mannose on Vero cells is shown in Figure 4. The cell viability was not significantly different between the treatments with poly-D-mannose of 0.125–8 mg/mL and untreated cells. Therefore, it can be concluded that poly-D-mannose shows no cytotoxicity.

#### 3.2.2. Hemolysis

The hemolysis results of poly-D-mannose are listed in Table 2. Regardless of whether deionized water or TritonX-100 was used as the positive control, the hemolysis rates of poly-D-mannose were all less than 5%. After incubation for 4 h, the hemolysis rate of poly-D-mannose was 0.50% ± 0.22% at 8 mg/mL both for deionized water and TriotonX-100. The hemolysis rates decreased to 0.26% ± 0.11% (positive control: deionized water) and 0.28% ± 0.12% (positive control: TriotonX-100) after treatment for 20 h. According to the judgment standard, the hemolysis rates of poly-D-mannose were all less 5% for 0.125–8 mg/mL, suggesting no hemolytic activity.

### 3.3. Biological Activity Analysis

#### 3.3.1. Anti-Biofilm Activity

The anti-biofilm activity of poly-D-mannose on the four pathogenic bacteria is shown in Figure 5. The biofilms of the four pathogens were inhibited to different degrees after treatment with poly-D-mannose. In the interval 0.25–2 mg/mL, poly-D-mannose showed a dose-dependent inhibitory effect on the biofilm formation of *S.* Typhimurium, but no significant influence on *E. coli* O157: H7, *S*. *aureus*, or *B*. *cereus*. The biofilms of *S.* Typhimurium and *E. coli* O157:H7 were reduced by 44% and 22%, respectively, at 0.25 mg/mL of poly-D-mannose. However, the poly-D-mannose had no effect on the biofilm formation of *S*. aureus and slightly promoted the biofilm formation of *B*. *cereus* at 0.25 mg/mL. With the increase in the concentration of poly-D-mannose, the formation of *S.* Typhimurium biofilm further decreased and biofilm formation was reduced by 68% at 2 mg/mL. For *E. coli* O157: H7, there was no significant difference in biofilm formation when the concentration of poly-D-mannose ranged from 0.5 to 2 mg/mL, with the reduction of 37%. The biofilms of *S*. *aureus* and *B*. *cereus* showed an inhibition effect when the concentration of poly-D-mannose was raised to 2 mg/mL, and the biofilm decreased by 34% and 47%, respectively.

#### 3.3.2. Antioxidant Activity

The antioxidant activity of poly-D-mannose was investigated by the capacity of radical scavenging in vitro, including of DPPH, hydroxyl radical, superoxide anion, and ABTS. The antioxidant activities of poly-D-mannose were analyzed in the concentration of 0.125–8 mg/mL, and the results are shown in Figure 6. The scavenging ability of poly-D-mannose varied for different free radicals, and exhibited the highest capacity on the DPPH among the four free radicals used in this study. With the increase in concentration, the scavenging ability of poly-D-mannose was significantly increased, and the maximum scavenging rate was 74% at 8 mg/mL. Poly-D-mannose also showed a certain scavenging effect on hydroxyl free radicals, with the maximum scavenging rate of 37%, which was significantly lower than that of DPPH. Poly-D-mannose showed slight scavenging activity on ABTS and superoxide radicals, with the scavenging rates of 16% and 10%, respectively, at 8 mg/mL.

#### 3.3.3. Anti-Inflammatory Result

Proliferation and NO production

The effect of poly-D-mannose on the proliferation of RAW 264.7 macrophages is shown in Figure 7a. As Figure 7a shows, the proliferation of macrophages treated with poly-D-mannose (0.125–2 mg/mL) in the presence of 1 µg/mL LPS was not significantly different from that of the cells only treated with 1 µg/mL LPS (positive control), suggesting no effect on the cell growth. Figure 7b shows the result of NO production. NO content of cells stimulated with LPS was increased significantly compared to that of the untreated cells. However, when macrophages were treated with poly-D-mannose in the presence of 1 µg/mL LPS, the NO production decreased in a dose-dependent manner. Compared to the positive control, NO decrease from 9.8 µM to 6.8 μM at 2 mg/mL of poly-D-mannose.

2.ROS production

Figure 7c,d show the results of ROS production in the macrophages treated with or without poly-D-mannose. There was almost no ROS production in the untreated macrophages according to the results of fluorescence spectrophotometry and using a fluorescence microscope. ROS can be obviously observed in LPS-treated cells, as shown in Figure 7c. After determination, it showed that poly-D-mannose had no effect on the ROS production at low concentrations (0.125 and 0.25 mg/mL), but inhibited the secretion of ROS at concentrations of 0.5–2 mg/mL, showing a significant difference compared to the LPS-stimulated cells. In addition, the result of fluorescence intensity also suggested that poly-D-mannose can significantly decrease the production of NO in macrophages at 1–2 mg/mL (Figure 7d).

3.The production of IL-6 and TNF-α

Two common cytokines (IL-6 and TNF-α) were preliminarily used for the anti-inflammatory evaluation of poly-D-mannose, and the results are shown in Figure 7e,f. There was almost no production of IL-6 and TNF-α in untreated cells (negative control). The TNF-α and IL-6 in LPS-treated macrophages were approximately 160 pg/mL and 700 pg/mL, respectively. After treatment with 2 mg/mL of poly-D-mannose, the production of TNF-α and IL-6 were decreased to 120 pg/mL and 600 pg/mL, respectively, showing potential anti-inflammatory effects.

## 4. Discussion

Mannan is an important functional polysaccharide, which can be obtained from various sources. Mannan from different sources expresses different properties, structures, molecular weights, and monosaccharide compositions. Plant mannan always has high viscosity, low water solubility, and wide molecular weight distribution, which ranges from tens of thousands to millions of Daltons, and the monosaccharide composition is not always homogeneous [1,43]. Yeast mannan exists in the outer layer of the yeast cell wall, which is composed of 90% mannose and 10% protein, and the molecular weight ranges from 20,000 to 200,000 Da [44,45]. Furthermore, the molecular weight of some mannans synthesized by engineering strains or enzyme catalysis was found to be thousands of Daltons [18,19]. In the present study, poly-D-mannose synthesized using the one-pot method [46] was composed of three larger components (>410,000 Da) and a smaller one with the molecular weight of 3884 Da. The branching degree was as high as 0.53. Compared with mannan prepared by other methods, the one-pot method was simpler, easer to repeat, and the production was purer. Furthermore, the present study shows that the poly-D-mannose is non-toxic and non-hemolytic, showing its potential application in the food, feed, and pharmaceutical industries.

Biofilm is an important survival mode of bacteria in nature [47]. Bacterial pathogens living in biofilm are more resistant to the external environment and more difficult to kill than planktonic bacteria. This results in long-term persistent infection, which causes serious harm to the medical and food industries and animal intestinal health [48]. Some previous studies have shown that natural polysaccharides, such as algal, plant, and animal polysaccharides, and bacterial exopolysaccharides, may also exhibit anti-biofilm activity on different Gram-positive and Gram-negative bacteria [15,49,50]. In our study, the poly-D-mannose synthesized by the one-pot method could inhibit the biofilm formation of four pathogens, which was reported for the first time. Among these, it obviously inhibited the biofilm of *S.* Typhimurium, with the highest inhibitory rate of 68% at the concentration of 2 mg/mL. For the other three pathogens, the inhibition ability of poly-D-mannose on the biofilm was weaker than that of *S.* Typhimurium and only showed a slight effect at high doses. The anti-biofilm activity of polysaccharide was not due to its bactericidal ability. Three main anti-biofilm mechanisms of polysaccharides have been reported, as follows: (i) acting as surface molecules to change the properties of biotic or abiotic substrates; (ii) acting as signal molecules to affect the expression of bacterial genes; and (iii) competitive inhibition of multivalent carbohydrate–protein interactions [51,52]. Previous reports have shown that *Fim* fimbriae in *Salmonella* are mannose-sensitive receptors. FimH is one of the structures of *Fim* fimbriae, and its main role is to specifically bind to mannose residues on the cell surface [53]. Poly-D-mannose in this study was composed of mannose. Therefore, we speculate that the FimH of *Salmonella* can be used as the binding site of poly-D-mannose, thus reducing the adhesion of *Salmonella*, which may be one of the reasons for its best inhibition capacity. Other possible mechanisms, such as whether poly-D-mannose influences biofilm formation as a surface molecule or signaling molecule, need to be further investigated.

Previous studies have shown that polysaccharides from different sources have strong antioxidant capacity and can be used as potential antioxidants to prevent oxidative damage in organisms [54]. Polysaccharide is a polyhydroxy compound with strong reducibility and can react with free radicals for degradation and structural transformation [55,56]. In our study, 8 mg/mL of poly-D-mannose showed the best scavenging ability against DPPH free radicals (74%), followed by hydroxyl free radicals (37%), but the scavenging ability against superoxide ion free radicals (16%) and ABTS free radicals (10%) was low. The scavenging activity of poly-D-mannose against these four free radicals was different from that of yeast mannan reported in other research. Liu et al. [11] isolated pure mannan from the yeast cell wall using the hot water method and evaluated its antioxidant capacity. The scavenging rates of both hydroxyl radicals and superoxide ion radicals were 50% at 3.2 mg/mL, which was higher than the activity of poly-D-mannose. Moreover, Faustino et al. [57] compared the scavenging effects of spray-dried yeast mannan (FDM), freeze-dried yeast mannan (SDM), and commercial mannan (CM) on ABTS and DPPH. At the concentration of 5 mg/mL, the scavenging rates of FDM and SDM on ABTS were 67% and 66%, respectively. However, the clearance rate of CM to ABTS was only 2%. The clearance rates of FDM and SDM for DPPH were low, at 13% and 19%, respectively, and only 8% for CM. Compared with the reported yeast mannan, poly-D-mannose showed better DPPH scavenging capacity, and the clearance rate reached 56% at 4 mg/mL. The scavenging activity of ABTS was higher than that of CM, reaching 9% at 4 mg/mL, but lower than that of FDM and SDM. The antioxidant properties of polysaccharides are closely related to the molecular weight, glycosidic bond type, and monosaccharide binding mode [58]. We reasonably suppose that the different sources and preparation methods lead to the molecular weight and structural differences between poly-D-mannose and other mannans, which resulted in different antioxidant results.

Inflammation is a natural protective response of the body to external stimuli such as pathogens, mechanical and chemical agents, and autoimmune responses [59,60]. However, long-term and chronic inflammation is harmful to the body, and results in a series of diseases including fever, asthma, atherosclerosis, arthritis, neurodegenerative diseases, and even cancer [61,62,63]. Therefore, it is essential to control the inflammatory response. Macrophages, as the first line of defense of the body’s immunity, are widely distributed and react in time. When activated, they will produce pro-inflammatory factors such as TNF-α, IL-6, and IL-1β [64]. RAW 264.7 macrophages are often used as a cell model to study the inflammatory response induced by LPS in vitro.

NO is one of the important inflammatory mediators. Excessive production of NO can lead to cytotoxicity, inflammatory reactions, and autoimmune diseases [65]. ROS can regulate molecular signaling pathways including MAPK and NF-κB pathways to increase cytokine expression. However, accumulation of ROS can cause damage to lipids, proteins, and DNA [66,67]. Therefore, it is necessary to balance the production of NO and reduce ROS. Our study suggested that poly-D-mannose can decrease the production of NO and ROS in LPS-treated cells in a dose-dependent manner. Macrophages stimulated by LPS can induce nitric oxide synthase (iNOS) to catalyze the production of NO [68]. Hence, it can be speculated that the expression of iNOS is inhibited after the addition of poly-D-mannose. In LPS-stimulated macrophages, the addition of poly-D-mannose reduced the content of ROS, which was more obvious at high concentrations. Meanwhile, it was also confirmed under fluorescence microscopy that the fluorescence intensity decreased in the presence of poly-D-mannose at concentrations of 1 mg/mL and 2 mg/mL, indicating the decrease in ROS content. After macrophages are activated by LPS, TNF-α is released, which further induces the production of IL-6 and IL-1β [69]. These three cytokines act on monocytes and macrophages to enhance the body’s immune function [70], but, similar to NO and ROS, excessive secretion also can cause great damage to the body [71]. Suppressing their overproduction can slow down inflammation. As shown in Figure 7e,f, the contents of TNF-α and IL-6 can be significantly decreased by poly-D-mannose at 0.5–2 mg/mL in a dose dependent manner. Inhibiting the production of pro-inflammatory cytokines by inhibiting the activation of NF-κB is an important anti-inflammatory mechanism of many polysaccharides [72,73]. Whether poly-D-mannose reduces TNF-α and IL-6 content by inhibiting the NF-κB-related signaling pathway needs further experimental verification.

Different sources of anti-inflammatory polysaccharides have been summarized in a review by Hou et al. [68]. The anti-inflammatory activity of poly-D-mannose was first evaluated in this study. Guo et al. [74] used Deoxynivalenol-stimulated porcine jejunum epithelial cell lines as cell models, and the addition of yeast mannan inhibited the expression of TNF-α, IL-8, and IL-6, but not that of IL-1β. Song et al. [75] reported the effect of Actigen, a second-generation mannan-rich fraction, on the intestinal tract of weaned piglets, and found that it could reduce the content of pro-inflammatory factor TNF-α and increase the content of anti-inflammatory factor IL-10 in serum. Wang et al. [76] found that the expression of Toll-like receptor 4, NF-κB, and IL-1β was inhibited in the ileum of broilers infected with *E. coli* when fed live yeast and mannose-oligosaccharides. In addition, glucomannan from plants also showed anti-inflammatory activity by regulating the balance of pro- and anti-inflammatory cytokines [77]. The anti-inflammatory activity of sugars is closely related to their structure, such as their monosaccharide compositions, molecular weights, chain conformations, and glycosidic linkages types and positions [68]. Previous studies reported that macrophages can express abundant carbohydrate receptors such as mannose receptors (MRs), which can specifically recognize saccharides rich in mannose. The presence of such recognition sites will send downstream signals and cause dynamic changes in the cell [78,79,80]. Poly-D-mannose synthesized in this study is composed of only mannose, so the anti-inflammatory activity may be related to it. In addition, the range of molecular weights of poly-D-mannose synthesized by the one-pot method in this study is relatively wide and it has a high degree of branching, which may also affect its anti-inflammatory activity.

## 5. Conclusions

In this study, a poly-D-mannose synthesized by the one-pot method had the characteristic absorption peaks of polysaccharide and consisted of mannose, with the total sugar content of 97.70%. Poly-D-mannose showed no cytotoxicity or hemolytic activity. In addition, it expressed antioxidant and anti-biofilm activities. Specifically, for the DPPH scavenging and the anti-biofilm activity on *S.* Typhimurium, the poly-D-mannose exhibited significantly different activities. In addition, poly-D-mannose reduced the secretion of NO and ROS in LPS-treated RAW264.7 macrophages, and decreased the contents of pro-inflammatory factors TNF-α and IL-6, indicating certain anti-inflammatory activity in vitro. Based on above results, the future application prospects of poly-D-mannose include the following: (1) As an antioxidant. It can be added to food as an antioxidant to preserve food flavor or added to cosmetics to reduce skin cell damage caused by oxidation. It can be used in livestock production to improve the antioxidative stress ability of livestock and poultry, thereby reducing the oxidative damage of tissue cells and preventing diseases. (2) As biofilm inhibitors. It can be used as a feed additive for chickens and other poultry to inhibit the biofilm formation of *S.* Typhimurium, thus reducing poultry diarrhea and improving production performance. (3) As an anti-inflammatory drug. It can be used for the treatment of human inflammatory diseases such as gastrointestinal inflammation, or added to feed to reduce the inflammatory response of livestock and poultry.

## Figures and Tables

**Figure 1 antioxidants-12-01579-f001:**
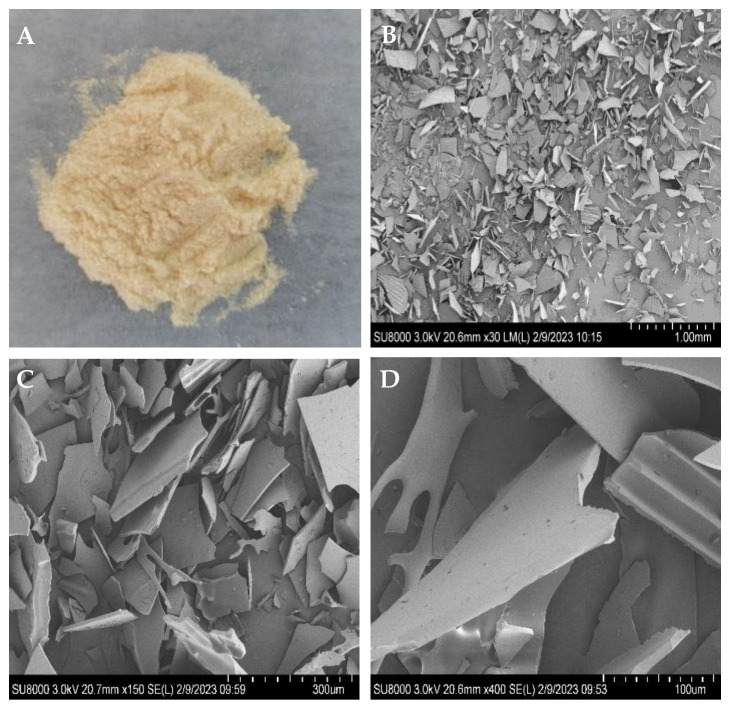
Macrostructure and microstructure of poly-D-mannose: (**A**): poly-D-mannose powder; (**B**): microstructure of poly-D-mannose at 30× magnification; (**C**): microstructure of poly-D-mannose at 150× magnification; (**D**): microstructure of poly-D-mannose at 400× magnification.

**Figure 2 antioxidants-12-01579-f002:**
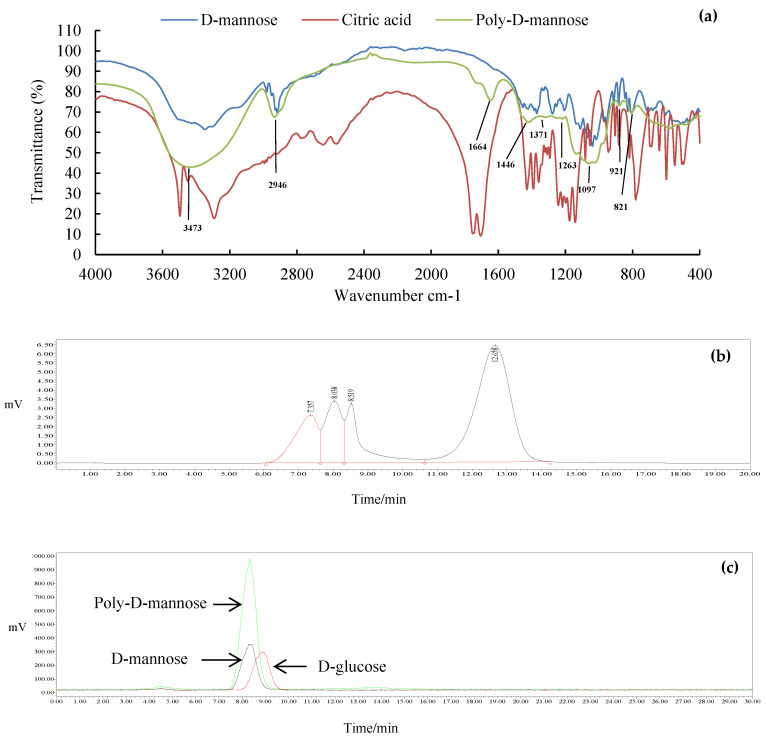
Results of FT-IR, molecular weight, and monosaccharide composition of poly-D-mannose synthesized by the one-pot method: (**a**): FT-IR comparison spectra of D-mannose, citric acid, and poly-D-mannose; (**b**): HPLC chromatogram of molecular weight distribution; (**c**): HPLC chromatogram of monosaccharide composition and monosaccharide standards.

**Figure 3 antioxidants-12-01579-f003:**
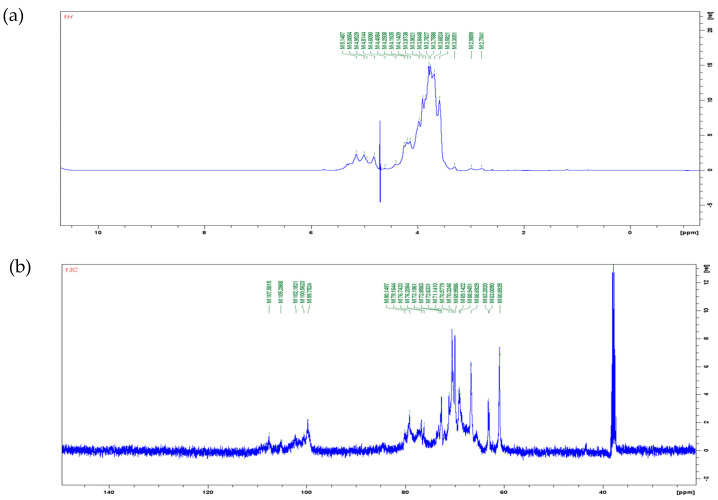
NMR spectra of poly-D-mannose (500 MHz, 298 K): (**a**): ^1^H NMR spectrum; (**b**): ^13^C NMR spectrum.

**Figure 4 antioxidants-12-01579-f004:**
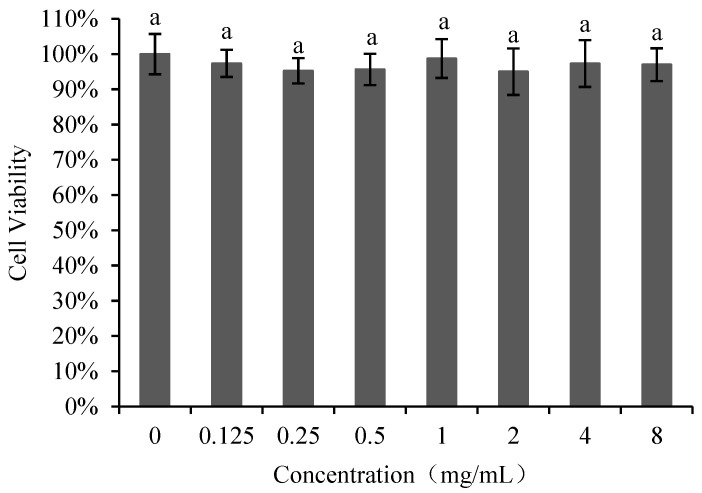
Cell viability of Vero cells after treatment with different concentrations of poly-D-mannose. Standard deviation of the samples (*n*= 3) is shown by vertical bars. ^a^ Significant differences (*p* < 0.05) are indicated with different lowercase letters.

**Figure 5 antioxidants-12-01579-f005:**
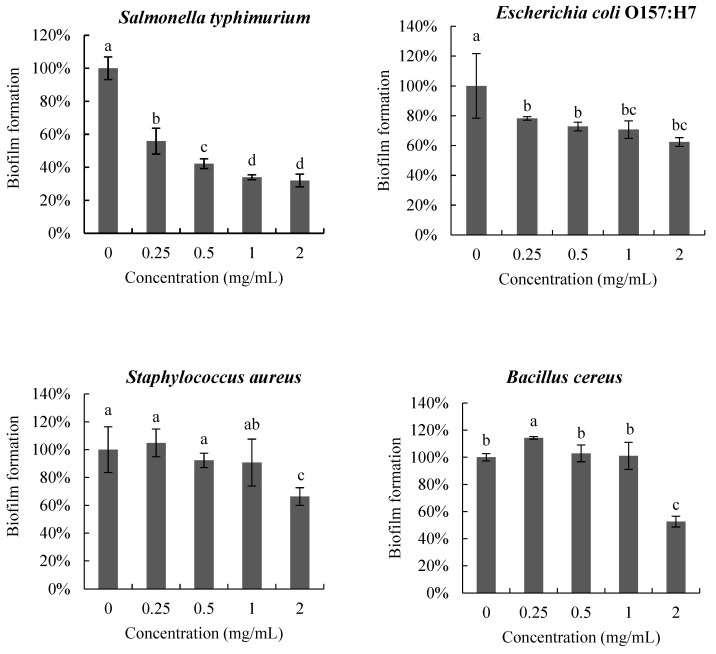
Anti-biofilm effects of poly-D-mannose on *Bacillus cereus*, *Staphylococcus aureus*, *Salmonella typhimurium*, and *Escherichia coli* O157: H7. Standard deviation of the samples (*n* = 3) is shown by vertical bars. ^a,b,c,d^ Significant differences (*p* < 0.05) are indicated with different lowercase letters.

**Figure 6 antioxidants-12-01579-f006:**
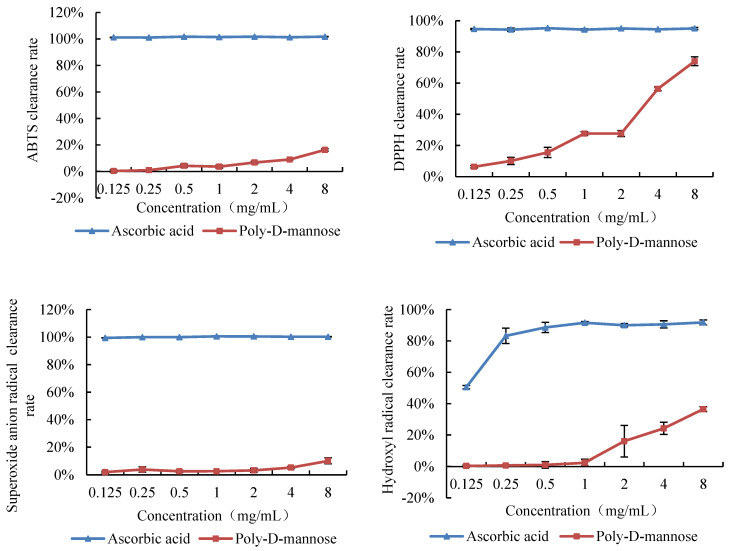
Scavenging activities of poly-D-mannose on ABTS, DPPH, superoxide anion radicals, and hydroxyl radicals. Standard deviation of the samples (*n* = 3) is shown by vertical bars.

**Figure 7 antioxidants-12-01579-f007:**
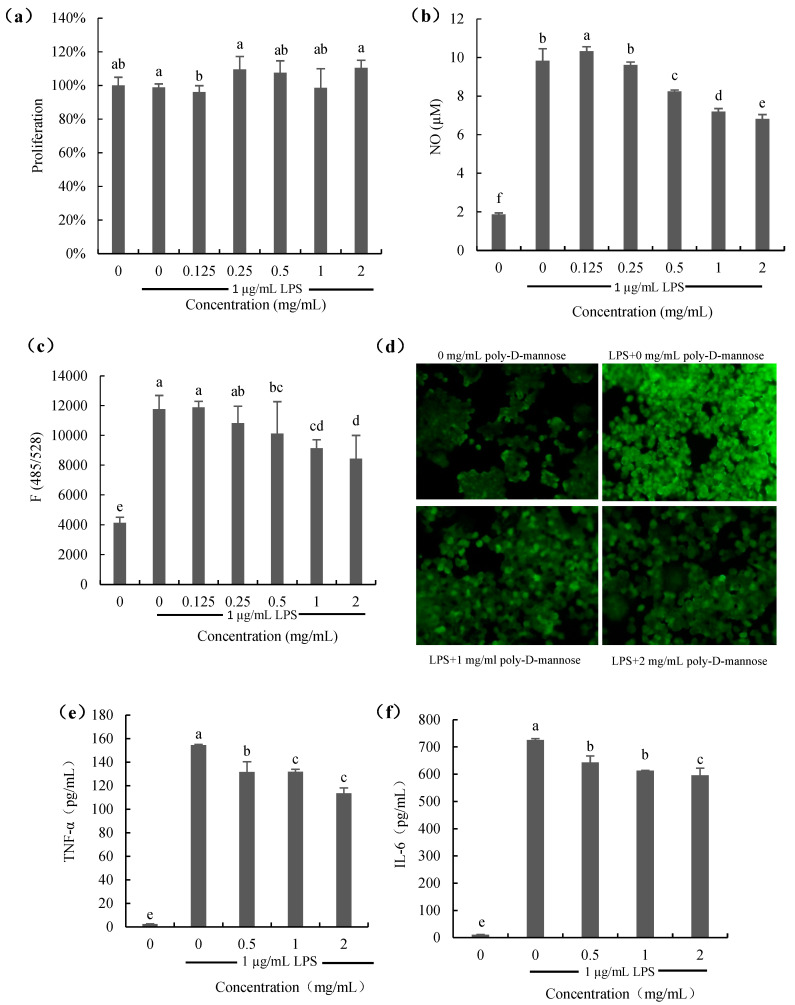
Effects of poly-D-mannose on RAW 264.7 macrophages: (**a**): proliferation of macrophages; (**b**): NO production; (**c**): ROS production; (**d**): ROS production observed by fluorescence microscopy (20×). (**e**): TNF-α production; (**f**): IL-6 production. Standard deviation of the samples (*n* = 3) is shown by vertical bars. ^a,b,c,d,e,f^ Significant differences (*p* < 0.05) are indicated with different lowercase letters.

**Table 1 antioxidants-12-01579-t001:** Types of sugar residues of poly-D-mannose.

	Peak Number	Derivatives	Linkage	Relative Molar Percentage
Terminal residues	1	2,3,4,6-Me_4_-Man	T-Man*p*-(1→	29.39
Linear residues	2	3,4,6-Me_3_-Man	→2)-Man*p*-(1→	6.34
	3	2,3,6-Me_3_-Man	→4)-Man*p*-(1→	11.24
	4	2,3,4-Me_3_-Man	→6)-Man*p*-(1→	22.72
	5	2,3,5-Me_3_-Man	→6)-Man*f*-(1→	6.60
Branch residues	6	2,3-Me_2_-Man	→4,6)-Man*p*-(1→	10.95
	7	3,4-Me_2_-Man	→2,6)-Man*p*-(1→	4.15
	8	6-Me-Man	→2,3,4)-Man*p*-(1→	0.91
	9	2,4-Me_2_-Man	→3,6)-Man*p*-(1→	1.95
	10	2-Me-Man	→3,4,6)-Man*p*-(1→	1.92
	11	3-Me-Man	→2,4,6)-Man*p*-(1→	3.83

**Table 2 antioxidants-12-01579-t002:** Hemolytic evaluation of poly-D-mannose at 4 h and 20 h; H_2_O and TritonX-100, respectively, are positive controls.

Hemolysis Rate of Different Concentrations of Poly-D-Mannose (mg/mL)								
		0.125	0.25	0.5	1	2	4	8
4 h	Pos1-H_2_O	0.00 ± 0.00%	0.00 ± 0.19%	−0.13 ± 0.11%	0.19 ± 0.00%	0.38 ± 0.19%	0.19 ± 0.19%	0.50 ± 0.22%
	Pos2-TritonX-100	0.00 ± 0.00%	0.00 ± 0.19%	−0.13 ± 0.11%	0.19 ± 0.00%	0.38 ± 0.19%	0.19 ± 0.19%	0.50 ± 0.22%
20 h	Pos1-H_2_O	0.06 ± 0.44%	−0.13 ± 0.11%	−0.26 ± 0.11%	−0.13 ± 0.11%	−0.06 ± 0.11%	−0.06 ± 0.11%	0.26 ± 0.11%
	Pos2-TritonX-100	0.07 ± 0.49%	0.14 ± 0.12%	0.28 ± 0.12%	0.14 ± 0.12%	0.07 ± 0.12%	0.07 ± 0.12%	0.28 ± 0.12%

The number before “±”: average hemolysis rate. The number after “±”: standard deviation of hemolysis rate.

## Data Availability

All methods and related data are presented in this paper. Additional inquiries should be addressed to the corresponding author.

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
