# Peer review of "A Poly-D-Mannose Synthesized by a One-Pot Method Exhibits Anti-Biofilm, Antioxidant, and Anti-Inflammatory Properties In Vitro"

_antioxidants, 2023, doi:10.3390/antiox12081579_

Round 1
Reviewer 1 Report
antioxidants-2484763-peer-review-v1
A novel poly-D-mannose synthesized by one-pot method exhibits anti-biofilm, antioxidant and anti-inflammatory properties in vitro
by Dandan Tian et al.
The authors report on the conversion of D-mannose to poly-D-mannose by thermal treatment in presence of citric acid. The resulting product is subjected to biological tests that show some interesting results in comparison to the starting material. The 2nd part is o. k. from my point of view but I do have severe doubts regarding the polymerization reaction. Maybe there are different expectations regarding the type of resulting polymer. Is it a polysaccharide where the repeating units are connected via glycosidic bonds or are the mannose units connected via an unknown linker? Citric acid is a common crosslinker. Therefore, oligomerization/polymerization via citric acid esters must be taken into account. It is throughout possible that this effect interferes with the linkage analysis. FTIT spectroscopy is nice but NMR spectroscopy would give more information on the structure of the product. SEC had been measured. Thus, the polymer should be soluble enough to acquire NMR spectra.
Specific comments:
L2: “poly-D-mannose” The “D” must be typed in small caps font (D in size of d; font size reduced by 3 points); This counts for the entire manuscript.
L11: The 1st sentence of the abstract can be omitted or moved to the introduction.
L15: …anti-inflammatory activity.
L21: …concentration range from 0.125 mg/mL to 8 mg/mL.
L22: “inhibition effect” instead of “inhibition rate” or similar
L79: are still existing
L84: Ref. 21 does not give information in synthesis
L88: Ref. 21 describes only enzymatic reactions
Section 2.2: Amounts of mannose and citric acid should be given in grams as well. What is the yield?
L122: samples were
L125: …was used for FT-IR spectroscopy…
L129: were measured as controls
Section 2.6: Was there a complete hydrolysis of the polymer prior to the sugar determination?
L157: specify “appropriate sodium hydroxide”
L163: specify “1mL sodium borohydride” I guess that a solution of a certain concentration had been used.
L165: specify “2 mL methanol solution”
L252: pH=8.0, 150 mM
L291: pH= 7.35
Section 3.1.1: What is the purpose of SEM in this regard? All samples have different history. Thus, different morphologies must be expected.
Section 3.1.2: The FTIR spectra should be stacked instead of being overlayed. The line thickness must be decreased. The spectrum of the polymer has a signal on the left hand side of the band at 1664 cm-1, which might be an ester bond. Moreover, the presence of glycosidic bonds is discussed based FTIR spectroscopy. This statement must be underpinned by NMR spectroscopy.
Section 3.1.3: A molar mass of 410.000 g/mol is stated, which corresponds to a degree of polymerization of 2.500. This is quite high for such kind of reaction. Figure 2b shows the elution curve but not the molar mass distribution. The given molar masses are questionable if the real structure of the polymer is not exactly known.
Fig. 5: What is “Vc”?
Section 4: The 1st part contains lot of information, which is state of the art. This should be reduced to a minimum here.
Author Response
Response to Reviewer 1 Comments
Point 1: Is it a polysaccharide where the repeating units are connected via glycosidic bonds or are the mannose units connected via an unknown linker? Citric acid is a common crosslinker. Therefore, oligomerization/polymerization via citric acid esters must be taken into account. It is throughout possible that this effect interferes with the linkage analysis.
Response 1: Thank you for your comment. In our study, The synthesis of poly-D-mannose referred to the preparation method of polydextrose, which were added in the manuscript [24,25]. D-mannose was used as monomer and citric acid as initiator to synthesize poly-D-mannose under high temperature and negative pressure. In the polymerization process, there was both esterification of mannose and citric acid and dehydration polymerization between monosaccharides (Ref:Chavan, R. S., Khedkar, C. D., Bhatt, S. Fat Replacer. Encyclopedia of Food and Health. 2016, 589-595.). Because the amount of citric acid added was much less than that of D-mannose, the polymer was dominated by monosaccharide units linked by glycosidic bonds, and contained small amount of citric acid and D-mannose binding units.
[24] Allingham, R. P. Polydextrose - A new food ingredient: Technical Aspects. In: Chemistry of Foods and Beverages: Recent Developments, Charalambous, G., Inglett, G.,Eds. Academic Press Inc: New York, 1982; pp 293-303.
[25] Younes, M., Aquilina, G., Castle, L., Engel, K. H., Fowler, P., Fürst, P., Gürtler, R., Gundert Remy, U., Husøy, T., Manco, M., Mennes, W., Moldeus, P., Passamonti, S., Shah, R., Waalkens Berendsen, D. H., Wölfle, D., Wright, M., Boon, P., Crebelli, R.,... Fernandez, M. J. F. Re‐evaluation of polydextrose (E 1200) as a food additive. EFSA Journal. 2021, 19(1).
Point 2: FTIR spectroscopy is nice but NMR spectroscopy would give more information on the structure of the product. SEC had been measured. Thus, the polymer should be soluble enough to acquire NMR spectra.
Response 2: Thank you for your comment. Since our synthetic poly-D-mannose is a mixture, we do not provide NMR spectra. However, according to your suggestion, 1H NMR and 13C spectrums of poly-D-mannose was added as shown in Figure 3 in the revised manuscript.
Point 3: L2: “poly-D-mannose” The “D” must be typed in small caps font (D in size of d; font size reduced by 3 points); This counts for the entire manuscript.
Response 3: Thank you for your kind suggestion. We have revised it in the manuscript.
Point 4: L11: The 1st sentence of the abstract can be omitted or moved to the introduction.
Response 4: Thank you for your good suggestion. The 1st sentence of the abstract have been deleted in the current version .
Point 5: L15: …anti-inflammatory activity.
Response 5: Thank you for your kind suggestion. We have made revisions in the manuscript based on your suggestion in the revised manuscript.
Point 6: L21: …concentration range from 0.125 mg/mL to 8 mg/mL.
Response 6: Thank you for your suggestion. We have revised it in the manuscript.
Point 7: L22: “inhibition effect” instead of “inhibition rate” or similar.
Response 7: Thank you for your good suggestion. We have deleted “with the inhibition rate of …” and made a few changes in this sentence.
Point 8: L79: are still existing.
Response 8: Thank you for your suggestion. We have revised it in the manuscript.
Point 9: L84: Ref. 21 does not give information in synthesis.
Response 9: Thank you for your comment. In Ref.21, “In general, HBPSs can be classified into synthetic, semisynthetic and natural ones” was described. We quote it in the manuscript. But there was indeed no further detailed description of the synthesis method. Therefore, we added relevant references as follows. We also added references to the manuscript.
[21] Kadokawa J., Tagaya H. Architecture of polysaccharides with specific structures: synthesis of hyperbranched polysaccharides. Polym Advan Technol. 2000, 11, 122-126.
[22] Satoh, T., Kakuchi, T. Synthesis of hyperbranched carbohydrate polymers by ring-opening multibranching polymerization of anhydro sugar. Macromol Biosci. 2007, 7(8), 999–1009.
Point 10: L88: Ref. 21 describes only enzymatic reactions.
Response 10: Thank you for your comment. I am sorry that I think you may be referring to Ref.22 in L88. For Ref. 22, we make the following explanation. One-pot method refers to the method of adding the reaction substrate together to the reaction vessel and preparing the reaction product through a one-step reaction. Ref.22 only provides examples of enzyme reaction methods, but the one pot method is not limited to enzyme catalysis, and other catalysts can also be selected as needed. We are only quoting from the literature on the advantages of the one-pot method, not referring to the enzymatic reaction method.
Point 11: Section 2.2: Amounts of mannose and citric acid should be given in grams as well. What is the yield?
Response 11: Thank you for your comment. The amounts of mannose and citric acid used in this study are 30 g and 0.9 g, respectively. We have added this data to the manuscript. The final yield of poly-D-mannose is 10 g.
Point 12: L122: samples were
Response 12: Thank you very much for your careful reading and kind suggestion. We have revised it in the manuscript.
Point 13: L125: …was used for FT-IR spectroscopy…
Response 13: Thank you for your good suggestion. We have revised it in the manuscript.
Point 14: L129: were measured as controls
Response 14: Thank you for your good suggestion. We have revised it in the manuscript.
Point 15: Section 2.6: Was there a complete hydrolysis of the polymer prior to the sugar determination?
Response 15: Thank you for your comment. Phenol-sulfuric acid method is a classic method for the determination of total sugars. By using this method, polysaccharides are first hydrolyzed into monosaccharides under the action of sulfuric acid, and rapidly dehydrated to form aldehyde derivatives, which then react with phenol to form orange yellow compounds. For your doubt, we haven't studied it before. If you think this method is not appropriate, we can delete this part.
Point 16: L157: specify “appropriate sodium hydroxide”
Response 16: Thank you for your good suggestion. The amount of sodium hydroxide is 80 mg. We have added this data in the manuscript.
Point 17: L163: specify “1mL sodium borohydride” I guess that a solution of a certain concentration had been used.
Response 17: Thank you for your comment. I am so sorry that we do not give a certain concentration. The concentration of “1mL sodium borohydride” is 4%. We have also added this data to the manuscript.
Point 18: L165: specify “2 mL methanol solution”
Response 18: Thank you for your comment. The mass fraction of methanol used in this study was greater than 99.5%.
Point 19: L252: pH=8.0, 150 mM
Response 19: Thank you for your suggestion. We have revised it in the manuscript.
Point 20: L291: pH= 7.35
Response 20: Thank you for your suggestion. We have revised it in the manuscript.
Point 21: Section 3.1.1: What is the purpose of SEM in this regard? All samples have different history. Thus, different morphologies must be expected.
Response 21: Thank you for your comment. SEM is a analysis method for observing the morphological properties of polysaccharides such as size, shape, and porosit. Different preparation methods will lead to the differences in the microstructure of polysaccharides. Because poly-D-mannose is prepared for the first time in our study, we wanted to use SEM to provide the microstructure of poly-D-mannose, which can be compared with mannan in other reported literatures.
Point 22: Section 3.1.2: The FTIR spectra should be stacked instead of being overlayed. The line thickness must be decreased. The spectrum of the polymer has a signal on the left hand side of the band at 1664 cm-1, which might be an ester bond. Moreover, the presence of glycosidic bonds is discussed based FTIR spectroscopy. This statement must be underpinned by NMR spectroscopy.
Response 22: Sincerely thank you for your suggestions. We have decreased the thickness of the lines in FT-IR spectra according to your suggestion. I am sorry that the band at 1664 cm-1 was ignored by us. I agree with you that this band is probably caused by the ester bond produced by the reaction of citric acid and D-mannose. Besides, we provided the 1H and 13C NMR spectroscopy to identify the type of glycoside bond in the revised manuscript (Figure 3).
Point 23: Section 3.1.3: A molar mass of 410.000 g/mol is stated, which corresponds to a degree of polymerization of 2.500. This is quite high for such kind of reaction. Figure 2b shows the elution curve but not the molar mass distribution. The given molar masses are questionable if the real structure of the polymer is not exactly known.
Response 23: Thank you for your comment. For your doubt, we explained it as follows: the preparation method of poly-D-mannose in this study referred to the preparation method of polydextrose as answered in “Point 1”. The difference from the synthesis method of polydextrose was that the monomer was different and sorbitol used as a plasticizer and chain terminator was not added. In the process of synthesis, D-mannose was randomly polymerized in the presence of citric acid to generate an irregular hyperbranched polymer with a wide range of molecular weights. Since chain terminator was not added, a large molecular weight could be produced. For the determination of poly-D-mannose molecular weight, we referred to the detection method of polysaccharide molecular weight [26,27]. Therefore, the results obtained based on the preparation and detection method are reliable.
The HPLC spectrum in Figure 2b is detected by GPC, and the molecular weight is calculated according to the standard curve drawn by dextran as the standard. The standard curve was showed in “section 2.5” in the manuscript.
[26] Zhou, C., Huang, Y., Chen, J., Chen, H., Wu, Q., Zhang, K., Li, D., Li, Y., & Chen, Y. Effects of high-pressure homogenization extraction on the physicochemical properties and antioxidant activity of large-leaf yellow tea polysaccharide conjugates. Process Biochem. 2022, 122, 87-94.
[27] Ma, Y., Xiu, W., Wang, X., Yu, S., Luo, Y., & Gu, X. Structural characterization and in vitro antioxidant and hypoglycemic activities of degraded polysaccharides from sweet corncob. J CEREAL SCI. 2022, 108, 103579.
Point 24: Fig. 5: What is “Vc”?
Response 24: Thank you for your comment. Vc refers to Ascorbic acid. We have replaced Vc with Ascorbic acid in Figure 5.
Point 25: Section 4: The 1st part contains lot of information, which is state of the art. This should be reduced to a minimum here.
Response 25: Thank you for your kind suggestion. We have integrated and reduced some sentences in the 1st part of Section 4.
Reviewer 2 Report
The study is interesting, but some information is lacking:
- Please include information regarding the analysis of replicate samples, whenever appropriate
- Regarding the graphical representations: please do not presente percentages above 100%, as it does not make since. I know it is due to standard deviations, but find a better way to present the results
- The authors must include a roadmap for future putative applications, the information already included in the conclusion section requires enhancement, it is rather basic and common to all research papers...
No comments
Author Response
Response to Reviewer 2 Comments
Point 1: Please include information regarding the analysis of replicate samples, whenever appropriate.
Response 1: Thank you very much for your good suggestion. We have conducted replicate samples for structural characterization and activity evaluation. The activity evaluation section has been elaborated in the methodology, but the structural characterization section lacks this information. We have supplemented it in the methodology section of the manuscript.
Point 2: Regarding the graphical representations: please do not presente percentages above 100%, as it does not make since. I know it is due to standard deviations, but find a better way to present the results.
Response 2: Thank you for your good suggestion. we tried our best to improve the expression of those data. However, I can’t find a more sutiable methods. As you know, it is because of standard deviations. So we think it is a resonable presence. In addition, some reported articles also used the similar expression as the following literature shown. So we sincerely wish you can agree with this expression. Or if you have some better methods, please send it to us. We are willing to imprve it according your suggestion.
Gu J., Zhang H., Yao H., Zhou J., Duan Y., & Ma H. Comparison of characterization, antioxidant and immunological activities of three polysaccharides from Sagittaria sagittifolia L. Carbohyd Polym. 2020, 235, 115939.
Wan C., Jiang H., Tang M., Zhou S., & Zhou T. Purification, physico-chemical properties and antioxidant activity of polysaccharides from Sargassum fusiforme by hydrogen peroxide/ascorbic acid-assisted extraction. Int J Biol Macromol. 2022, 223, 490-499.
Point 3: The authors must include a roadmap for future putative applications, the information already included in the conclusion section requires enhancement, it is rather basic and common to all research papers...
Response 3: Thank you very much for your kind suggestion. Based on the data available to us, We have added the detailed description of poly-D-mannose applications for the future in the section of “Conclusion”
Reviewer 3 Report
Manuscript deals with the synthesis of poly-D-mannose by one-pot method under the condition of high temperature and negative pressure using D-mannose. Molecular characteristics, as weight, degree of branching, monosaccharide composition and total sugar content has been determined, and the characteristic groups were analyzed by Fourier transform infrared spectroscopy (FT-IR). The safety of poly-D-mannose, including cytotoxicity and hemolysis, was also evaluated. Relevant experiments on its functional activities such as anti-biofilm, antioxidant and immune/anti-inflammatory activities were carried out.
Methods are perfectly and exhaustively described, and they are adequate.
Results, discussion, and conclusions are well documented, and they are sound.
I only have minor concerns
line 436.- During 0.25-2 mg/mL must be "In the interval 0.25 - 2 mg/ml"
Table 2 must be improved, and meaning of different percentages should be indicated in a footnote.
3.1. Subsectio on line 454 must be removed.
The meaning of Vc line on figure 5 must be indicated in the footnote
Minor editing of English language required
Author Response
Response to Reviewer 3 Comments
Point 1: line 436.- During 0.25-2 mg/mL must be "In the interval 0.25 - 2 mg/ml"
Response 1: Thank you very much for your good suggestion. We have made revisions according to your suggestion in the present version of manuscript.
Point 2: Table 2 must be improved, and meaning of different percentages should be indicated in a footnote.
Response 2: Thank you for your kind suggestion. We have improved Table 2 and the meaning of different percentages was indicated in a footnote.
Point 3: 3.1. Subsectio on line 454 must be removed.
Response 3: Thank you for your kind suggestion. We have deleted “3.1. Subsectio”.
Point 4: The meaning of Vc line on figure 5 must be indicated in the footnote
Response 4: Thank you for your good suggestion. Vc refers to Ascorbic acid. We have replaced Vc with Ascorbic acid in Figure 5.
Round 2
Reviewer 1 Report
Thanks a lot for considering my comments. However, the term "poly-D-mannose" is still misleading because the chemical structure of this condensation product is not fully understood. Therefore, SEC-elution curves can be shown in a qualitative manner. However, calibration with dextran (a very uniform polymer) should be omitted here.